# The Effect of the Housing System, Season and the Linseed Oil Ethyl Esters Additive on Selected Blood Parameters in Rabbits

**DOI:** 10.3390/ani12202773

**Published:** 2022-10-14

**Authors:** Katarzyna Roman, Martyna Wilk, Piotr Książek, Katarzyna Czyż, Adam Roman

**Affiliations:** 1Department of Animal Nutrition and Feed Science, Wroclaw University of Environmental and Life Sciences, 51-630 Wroclaw, Poland; 2Independent Researcher, 51-649 Wroclaw, Poland; 3Division of Sheep and Fur Animals Breeding, Wroclaw University of Environmental and Life Sciences, 51-630 Wroclaw, Poland; 4Department of Environment Hygiene and Animal Welfare, Wroclaw University of Environmental and Life Sciences, 51-630 Wroclaw, Poland

**Keywords:** rabbit, ethyl esters linseed oil, fatty acids, serum blood, erythrocytes

## Abstract

**Simple Summary:**

Evaluation of rabbit hematology parameters is important in terms of assessing the physiological state of animal organisms. However, most reference ranges originate from experimental laboratory studies. Therefore, rabbit hematology results are difficult to interpret. The aim of the study was to show the impact of different factors: environmental conditions and season on selected rabbit blood parameters. Moreover, many studies confirm the positive effect of polyunsaturated fatty acids on the health of various animal species. Therefore, another goal of the present work was to evaluate the effect of a feed additive, i.e., esterified linseed oil, on selected rabbit blood parameters. The experiment was carried out with the use of two experimental factors: animal keeping conditions (laboratory and outdoor) and the season (summer and winter), with a control and experimental group. The animals kept in the experimental group received an addition of esterified linseed oil to the feed for a duration of 2 months. The blood samples were collected at the beginning of the experiment, as well as at the end of treatment. Moreover, blood samples were collected also after two months from the end of the experiment. The data presented in this study report the influence of the environmental conditions, season and ethyl linseed oil supplementation on the selected blood parameters, especially the profile of unsaturated fatty acids. These results show the influence of ethyl esters of linseed oil additive and extend the database of hematological blood parameters in rabbits, as a result of which it will contribute to improvements in laboratory diagnostics.

**Abstract:**

The composition of the diet, including the amount and type of lipid supplements, influences the products formed in the digestive tract, their levels in the blood and their deposition in the tissues. One example of a plant rich in polyunsaturated fatty acids is flax (*Linum usitatissimum*). The aim of the presented work was to evaluate the effect of the linseed oil ethyl esters additive and environmental conditions on the selected rabbit blood parameters. The study included two types of animal maintenance (laboratory conditions and external conditions) and two seasons (summer and winter), a total of four study stages. At each stage, a control group and an experimental group were specified. The feed supplement in the form of esterified linseed oil was administered to the experimental animals for two months. The blood samples were collected at the beginning of the experiment, as well as at the end of treatment. Moreover, blood samples were collected also after two months from the end of the experiment. During the experiment, morphological and biochemical parameters of the rabbits’ blood were examined. The results of the content of fatty acids were determined in the erythrocytes and serum blood samples and grouped into saturated and unsaturated fatty acids, especially omega-3 and omega-6 fatty acids. In an internal study, the influence of housing conditions and season on selected morphological and biochemical parameters of rabbit blood was confirmed. Furthermore, expected beneficial changes in the fatty acid profile in erythrocytes and blood serum were observed as a result of supplementation. A significant increase in omega-3 fatty acids was noted as well as a substantial decrease in the ratio of omega-6 to omega-3 fatty acids (*p* < 0.01). Most importantly, the linseed oil ethyl ester supplement used did not adversely affect the health of the rabbits, as evidenced by biochemical and blood morphological indices remaining within, known so far, reference limits or showing only slight fluctuations (*p* > 0.05). However, the obtained results extend the database of hematological blood parameters in rabbits, thus, contributing to improvements in laboratory diagnostics for fur animals.

## 1. Introduction

The European rabbit (*O. cuniculus*) occurs on the European continent in three forms: wild, feral and domestic [1]. Domestic rabbits (*Oryctolagus cuniculus* f. *domestica*) belong to the earliest domesticated fur animals in the world. The domestication of European rabbits began with keeping them in separated and fenced areas, in a semi-wild state [2,3]. The familiar domestic form is typified by a great variety of breeds and strains, which are used for meat, fancy and laboratory animal production [1,4].

The success in rearing and breeding rabbits is established by many factors, such as consumer preferences, dietary trends, climate, nutrition or feed quality. For proper growth, rabbits as herbivores require green feeds, root crops and full-value concentrated feeds. Fat, as one of the main nutrients, supplies two groups of fatty acids: saturated and unsaturated fatty acids (SFAs and UFAs). A part of UFAs, acids known as polyunsaturated fatty acids (PUFAs) can be divided into omega-3 (n-3) and omega-6 (n-3) acids. In the n-3 fatty acids group, extremely valuable α-linolenic acid (ALA) is distinguished, with its derivatives eicosapentaenoic acid (EPA) and docosahexaenoic acid (DHA), and in n-6 group linoleic acid (LA) and its derivatives. Polyunsaturated fatty acids play an important role in the functioning of the animal and human body [5,6]. The composition of the diet, including the amount and type of lipid supplements, influences the products formed in the digestive tract, their absorption and their deposition in the tissues, in the hair coat and in the milk [7,8]. Animal products can be a source of health-promoting ingredients, e.g., omega-3 fatty acids, which have a beneficial effect on the health of the consumer. The most important effects of n-3 fatty acids include, among others: anti-atherosclerotic action, reduction in cholesterol concentration, normalization of blood pressure, inhibition of the development of ischaemic heart disease and coronary heart disease, anti-inflammatory and anti-allergic action, inhibiting of excessive immune response, as well as the acute course of inflammatory processes in viral and bacterial etiology, protection of the immune system [9]. In addition, n-3 fatty acids show an anticoagulant effect due to the prolongation of bleeding time by reducing the susceptibility of platelets to clumping due to the inhibition of the formation of highly prothrombotic substances, anticancer effects and beneficial effects on the skin and hair, as well as therapeutic effects in the case of skin disorders (e.g., atopic dermatitis) [6].

Moreover, dietary supplementation of n-3 fatty acids improved the fertility rate and number of weaned rabbits [10]. PUFA supplementation reduced kit mortality at the second parturition and improved the kit’s weight and length compared to kittens from the control group [11]. The use of rapeseed and fish oil, as the source of PUFAs, in the diet had a beneficial effect on rabbits’ final body weight and did not affect the level of intramuscular fat in the longissimus dorsi muscle [12]. The chemical composition of the ration, including the amount and type of lipid supplements (vegetable oils or oilseeds) or feed additives, affects the lipid profile of animal products [7,13,14,15]. The main plant sources of n-3 acids are nuts and vegetable oils [16]. One extremely rich source of valuable fatty acids is linseed [17]. In flax seeds, more than 80% of the total fatty acids are PUFAs, mainly α-linolenic acid. Therefore, plants rich in PUFAs are widely used in the diet of humans and animals to enrich it with valuable α-linolenic acid. Our previous study indicates that addition of omega-3 fatty acid source (linseed oil ethyl esters) has a favorable effect on rabbit coat [8]. Flax seed has also been used in pets food, e.g., dogs, to improve the quality of fur [18]. Despite the many beneficial nutritional properties, linseed also contains anti-nutrients—linamarin and linase. Anti-nutrients found in many plants, including linseed, can be reduced by thermal, hydrothermal or chemical treatment, e.g., esterification process [18]. Extremely valuable α-linolenic and linoleic acids are not biosynthesized in the animal body; consequently, it is necessary to supply these acids with food [19,20].

The environmental conditions (e.g., production system) had no significant effect on the content of most fatty acids, which was confirmed by Daszkiewicz et al. [21]. However, various external factors may influence the breeding profitability parameters, reproduction indicators, immunity and health status and quality of animal products. For example, because of a lack of sweat glands, thermoregulation is extremely poor in rabbits, which is why heat stress adversely affects welfare, feed consumption and utilization, immunity and health status, growth, reproduction, and milk production in rabbits [22].

Analysis of morphological and biochemical parameters of blood is an important part of assessing the physiological, nutritional and pathological status of an animal. Blood morphological parameters can change under the influence of various external factors or physiological changes in the body, e.g., pregnancy [23]. It is also worth controlling polyunsaturated fatty acid levels.

The aim of the study was evaluating the influence of environmental conditions (season and maintenance conditions) on selected blood parameters in termond white rabbits. In addition, the second goal of the experiment was to evaluate the effect of a feed supplement, in the form of linseed oil ethyl esters, on selected rabbit blood parameters, including the fatty acid profile.

## 2. Materials and Methods

### 2.1. Experimental Design

The study was conducted at Wroclaw University of Environmental and Life Sciences (Poland), on white termond rabbits. The research project detailing the experimental factors, such as animal maintenance (laboratory conditions and external conditions) and two seasons (summer and winter) have been described in greater detail in an earlier work, which is part of one research project focusing on the assessment of the usefulness of linseed oil ethyl esters in rabbit nutrition [8]. The experiment was divided into four stages (I, II, III, IV), sequentially: I-L-S (I-Laboratory—Summer), II-L-W (II-Laboratory—Winter), III-O-S (III-Outdoor—Summer) and IV-O-W (IV-Outdoor—Winter), which lasted 16 weeks.

### 2.2. Nutrition

During the experiment, 16 termond rabbits were divided into two groups: control (C—without addition), experimental (E—with addition of linseed oil ethyl esters), with 8 replications in each. Rabbits were fed in accordance with the requirements of reproductive rabbits [24,25]. Detailed information about the method of feeding, preparation and administration of linseed oil ethyl esters is presented in the aforementioned work of our team [8]. Feed ingredients and nutrient ratio of granules are presented in Table 1 (according to previous research [8]). Ethyl esters of polyunsaturated fatty acids obtained (directly in the laboratory, every 3 weeks) from cold-pressed linseed oil were used [26]. Fatty acid profiles of the granulate, hay and linseed oil ethyl esters are presented in Table 2 (according to previous research [8]). The addition of ethyl esters was to bring the n-6: n-3 ratio to a value of about 1:1 (Table 3, according to previous research [8]), the most favorable for the conversion of α-linolenic acid to EPA and DHA [27]. Before starting the research, the experimental addition was tested on 10 rabbits not covered by the experiment and harmful symptoms were not noted.

### 2.3. Analysis of Selected Blood Parameters

During the experiment, the morphology and biochemical tests of rabbit blood were carried out in order to monitor the animals’ health. The morphological composition of the collected material was determined including the following parameters: white blood cell count (WBC), lymphocytes (Lym), monocytes (Mono), granulocytes (Gran), red blood cell count (RBC), hemoglobin concentration (HGB), hematocrit (HCT), mean red cell volume (MCV), mean hemoglobin weight per cell (MCH), mean hemoglobin concentration per cell (MCHC), Red Cell Distribution Width (RDW), and platelet count (PLT) using the Abbott Cell-dyn 3700 Hematology Analyzers (Abbott Diagnostics, Abbott Park, IL, USA). Further, selected biochemical parameters such as alanine aminotransferase (ALT), aspartate aminotransferase (AST), fibrinogen and urea, were determined using an Olympus AU400 Chemistry Analyzer (Olympus and Beckman Coulter, Brea, CA, USA). Blood samples were taken from the rabbits before the start of the study, 8 weeks after administration of the preparation and from the end of supplementation. Material for the study was obtained from the middle ear artery of rabbits, immediately cooled to 4 ℃ and transported to the laboratory.

### 2.4. Fatty Acids Profile

Blood samples for the analysis of the fatty acid profile, as in the case of morphological analysis, were taken three times. Fat from serum and morphotic elements was extracted using the Folch method and fatty acid methyl esters were obtained using the Christopherson-Glass method [28,29]. The determination of saturated and unsaturated fatty acids was conducted using a gas chromatograph with an FID detector (7890A, Agilent Technologies, Santa Clara, CA, USA). In addition, levels of individual omega-3 (n-3) fatty acids were determined: α-linolenic acid (ALA), stearidonic acid (OTA), eicosapentaenoic acid (EPA), osmondic acid (DPA), docosahexaenoic acid (DHA) and omega-6 (n-6) fatty acids: linoleic acid (LA), γ-linolenic acid (GLA), citric acid (EDA), dihomo-γ-linolenic acid (DGLA), arachidonic acid (AA), and docosatetraenoic acid (DTA).

### 2.5. Statistical Analysis

Three-way analysis of variance (ANOVA) was used to analyze the data for main effects: housing conditions, season and linseed oil ethyl ester supplementation [30] Duncan’s multiple range test was used to confirm significant differences between the groups. The significance of differences was presented on two levels: *p* < 0.01 (upper case letters—A, B) and *p* < 0.05 (lower case letters—a, b).

## 3. Results

### 3.1. Morphology of Rabbit Blood

The addition of linseed oil ethyl esters did not affect (*p* > 0.05) the morphological parameters of blood collected from rabbits (Table 4). However, both the maintenance conditions and the season changed the blood morphological parameters of rabbits. The living conditions of the rabbits significantly influenced the content of monocytes (0.47 × 109/L and 0.32 × 109/L), MCH (19.78 and 22.28 pg; *p* < 0.01), lymphocytes (3.34 and 4.25 × 109/L), MCV (59.10 and 61.08 fL) and MCHC (334.84 and 341.90 g/L; *p* < 0.05), respectively, in animals kept in laboratory and outdoor cages. The blood of rabbits kept in external conditions was characterized by higher values of those parameters compared to the blood of rabbits kept in laboratory conditions, while the season significantly influenced the content of RBC, HGB, HCT and MCV (*p* < 0.01), as well as WBC, monocytes (*p* < 0.05). Lower levels of WBC, monocytes, as well as lower MCV were noted in the blood of rabbits kept during the winter. On the other hand, higher levels of RBC, HGB and HCT were noted in the blood of rabbits kept over the winter.

### 3.2. Biochemical Parameters of Rabbit Blood

The addition of linseed oil ethyl esters affected the urea level of blood collected from rabbits (Table 5). The urea level in the blood obtained from the experimental animals was statistically lower (*p* = 0.0421) compared to the blood of the control. The maintenance conditions also influenced the urea level (*p* = 0.0078) and fibrinogen level (*p* = 0.0025) of rabbit blood, which were lower in laboratory conditions. The season did not affect the biochemical parameters of blood collected from rabbits.

### 3.3. Fatty Acid Profile of Rabbit Blood

The addition of linseed oil ethyl esters significantly (*p* < 0.01) affected the fatty acid profile of rabbit blood erythrocytes (Table 6). Blood obtained from rabbits in the experimental group showed higher levels of unsaturated fatty acids, including MUFA, PUFA and omega-3 and omega-6 acids, compared to blood obtained from animals in the control group. The addition of linseed oil ethyl esters reduced saturated fatty acid levels and the n-6/n-3 ratio (*p* < 0.01). Statistical analysis of the fatty acid profile showed a significant effect (*p* = 0.0003) of housing conditions on SFA levels, which was lower in animals kept under outdoor conditions.

The impact of the major factors on the levels of individual fatty acids of the omega-3 and omega-6 family in rabbit blood erythrocytes is shown in Table 7. Statistical analysis showed a significant effect (*p* < 0.01) of the supplementation of linseed oil ethyl esters on the rise in the levels of omega-3 fatty acids: ALA, DHA, EPA and omega-6 fatty acids: LA, GLA and AA. The fatty acid profile of blood erythrocytes from rabbits maintained under outdoor conditions was characterized by lower levels of ALA (*p* < 0.05), AA and DTA (*p* < 0.01). On the other hand, the levels of GLA, EDA and DGLA acids were significantly higher (*p* < 0.01) in erythrocytes of blood sampled from animals kept under outdoor conditions. Blood erythrocytes obtained from animals kept in summer had higher levels of DPA (*p* = 0.0143) and GLA (*p* = 0.0000) compared to blood erythrocytes from rabbits kept in winter. Furthermore, the blood erythrocytes of rabbits kept during the summer period had lower DGLA levels (*p* = 0.0028) compared to rabbits kept in the winter period.

The effects of the main experimental factors on the serum fatty acid profile of rabbits are shown in Table 8. The addition of linseed oil ethyl esters increased the levels of UFAs, PUFAs, omega-3 and omega-6 acids and decreased the n-6/n-3 ratio. Blood serum obtained from rabbits kept under laboratory conditions had significantly lower MUFA levels (*p* = 0.0091), compared to blood serum obtained from animals kept under outdoor conditions. Animals kept in summer had higher serum levels of n-6 acids (*p* = 0.0045) compared to animals kept in winter.

The effects of the main factors on serum levels of individual omega-3 and omega-6 fatty acids in rabbits are shown in Table 9. Statistical analysis showed a significant effect of the addition of linseed oil ethyl esters on both the increase in the levels of omega-3 fatty acids: ALA (*p* < 0.05) and DHA and EPA (*p* < 0.01) and the levels of omega-6 fatty acids: LA (*p* < 0.05) and EDA, DGLA and DTA (*p* < 0.01). The serum fatty acid profile of rabbits maintained under outdoor conditions was characterized by higher levels of LA, GLA (*p* < 0.05) and DGLA (*p* < 0.01) compared to serum from animals maintained under laboratory conditions. In turn, the serum fatty acid profile of rabbits kept under laboratory conditions was characterized by higher levels of AA (*p* < 0.01) compared to serum from animals kept under outdoor conditions. Statistical analysis of the results also indicated the effect of season on the differences in serum levels of OTA and DTA (*p* < 0.05) and LA and AA (*p* < 0.01) in rabbits.

## 4. Discussion

Environmental stressors in the rabbit industry lead to a deterioration in rabbits’ health, e.g., organ damage, oxidative stress, disordered endocrine regulation, suppressed immune function and reproductive disorders, which cause decreased production performance [31]. Analysis of morphological and biochemical parameters of blood is an important part of assessing the physiological, nutritional and pathological status of an animal. However, most hematology reference ranges come from studies conducted on rabbits of the same age and breed, as well as kept in the same environmental conditions, which is why rabbit hematology results are difficult to interpret. As reported by Melillo [32], many texts amalgamate references and create wide ranges, which include almost any result. The author also points out that healthy pet rabbits are hard to find and samples from acute-condition rabbits may show changes due to malnutrition or improper husbandry.

Various physiological, nutritional and external factors may influence the breeding profitability parameters, reproduction indicators, immunity and health status and quality of animal products. Harcourt-Brown and Baker [33] reported that cage-kept rabbits, fed on commercial feed and struggling with dental disease had lower HCT, RBC counts, HGB values and lymphocyte counts in comparison with rabbits kept outside with a more natural diet and exercise. The primary role of lymphocytes is to respond to activities stimulating the immune system. The total white blood cell count can be used to characterize chronic stress caused by malnutrition, inappropriate husbandry, prolonged social stress or dental disease. In rabbits, increased adrenaline levels (acute stress) have been shown to induce lymphocytosis, while increased cortisol levels (chronic stress) lead to lymphopenia [32,34]. In an internal study, lower lymphocyte values in animals kept in laboratory conditions compared to animals kept outdoors were observed; however, in the study mentioned, these animals were kept in cages. Lower lymphocyte values (acute stress) in rabbits kept under laboratory conditions may be due to environmental stress (e.g., the loud sounds reflected back off the walls). Higher leukocyte values in animals kept during the summertime (chronic stress) may be caused by slight heat stress, due to the poor thermoregulation in rabbits [22]. However, the experiment was conducted under temperate climate conditions, which is characterized by lower summer temperatures compared to a tropical climate; therefore, it is suspected that despite a significant increase in leukocyte values, the animals experienced no heat stress. Moreover, despite statistically significant differences, leukocyte and lymphocyte values were within the reference range. This confirms that rabbits kept during summertime were not exposed to heat stress, which could adversely affect rabbit immunity and health status [22].

Rabbits under 12 weeks of age showed lower white blood cell and red blood cell counts [32]. However, the animals used in the experiment were 13–16 weeks old, so the changes shown are not due to the age of the animals. The reference range for HCT in rabbits is between 30% and 50%; values higher than 45% may indicate dehydration [35]. Low red blood cell and hemoglobin levels combined with HCT values below 30% would indicate anemia. However, in the experiment conducted, all groups of animals presented normal values for these parameters.

Determining the levels of biochemical blood parameters is important in terms of assessing the physiological state of animal organisms. Highly important biochemical blood parameters are ALT and AST, which make it possible to diagnose diseases of various internal organs. Due to its specificity for liver tissue, ALT is a useful indicator of liver cell damage. Slightly high levels of ALT may be caused by contact with toxic substances in low concentrations (e.g., sawdust resins, food aflatoxins) [36]. An increase in blood enzyme activity has been noted in environmental stress situations. Nakyinsige et al. [37] linked the increase in ALT and AST with heat stress, which occurred during transport of rabbits (independently of the duration) under hot humid tropical conditions. Aspartate aminotransferase (AST) increases also after physical immobilization of rabbits, especially if they are not familiar with being handled. In the tests carried out, ALT and AST levels were within the reference range in all groups and stages of the experiment, which allows us to conclude that both factors, season and housing system, did not cause environmental stress in rabbits. Fibrinogen, as an indicator of blood coagulation and acute phase protein, indicates ongoing inflammation in the body. In turn, urea is an indicator of kidney function. Blood urea can be an indicator of liver failure or muscle mass losses and this may be indicated by understated values of this parameter. Excessively high levels may indicate a high-protein diet [32].

It is well known that the amount and type of lipid supplements (vegetable oils or oilseeds) in the ration affects the blood lipid profile (which was confirmed in the presented study), thus, the lipid profile of animal products. Current trends in the production of pro-health foods are forcing scientists to look for methods to modify the fatty acid profile of animal products [7,13,14,15,38,39,40,41]. On the other hand, it is worth paying attention to the beneficial effect of omega-3 acids, in particular ALA, as a precursor to EPA and DHA, on the health of farm animals. Rabbits may constitute a research model for the study of the influence of omega-3 fatty acids on animal health. The linseed oil ethyl ester supplement used did not adversely affect the health of the rabbits, as evidenced by the values of individual blood biochemical indices remaining within reference limits or showing only slight fluctuations.

Mattioli et al. [10], administering extruded flaxseed to rabbits in a feed ration, recorded higher levels of ALA, EPA and DHA acids in the blood, compared to the control group. In our study, an increase in LA, GLA and AA in the fatty acid profile of blood erythrocytes and EDA, DGLA and DTA in the fatty acid profile of blood serum was also observed as a result of linseed oil ethyl ester supplementation. Marounek et al. [42] reported that with increasing levels of dietary supplementation of unsaturated acid (CLA) to rabbits, the concentration of CLA in tissue lipids increased. Hadjadj et al. [43] noted correlations between blood fatty acid levels and ovulation rate and embryonic development. Mattioli et al. [10] reported that enrichment of the diet with n-3 fatty acids can be considered a good strategy to improve reproductive performance in rabbits. Therefore, erythrocytes can serve as a model system for studies on fatty acid supplementation in relation to other body cells as well as culture indicators. In contrast, the results for the fatty acid profile obtained from blood serum are not very stable and reflect the immediate, temporary effect of supplementation rather than its long-term effect [44].

## 5. Conclusions

In the presented study, the influence of housing conditions and season on selected morphological and biochemical parameters of rabbit blood was confirmed. Despite the statistical differences, morphological and biochemical parameters of rabbit blood were within the reference range in all groups and stages of the experiment, which allows us to conclude that both factors, season and housing system, did not cause environmental stress in rabbits. Supplementation with esterified linseed oil had a positive effect on the fatty acid profile in erythrocytes and blood serum; the level of omega-3 fatty acids increased and the n-6: n-3 ratio decreased. The linseed oil ethyl ester supplement used did not adversely affect the health of the rabbits, as evidenced by biochemical and blood morphological indices remaining within reference limits or showing only slight fluctuations. Research should continue into the use of the linseed oil ethyl ester preparation in rabbits kept for meat production, as analysis of the animals’ blood shows its positive effects on the fatty acid profile, especially a significant increase in the levels of valuable unsaturated acids, including those in the omega-3 and omega-6 groups.

## Figures and Tables

**Table 1 animals-12-02773-t001:** Ingredients (g/kg) and chemical composition of granules (g/kg of dry matter) [8].

Ingredients		Composition	
Alfalfa	205	Crude protein	166.3
Grass mixture	135	Crude fiber	148.2
Wheat bran	230	Crude fat	21.4
Dried molasses beet pulp	120	Crude ash	84.2
Beet molasses	100	Calcium	11.2
Sunflower post-extraction meal	60	Sodium	2.6
Rapeseed post-extraction meal	50	Phosphorus	9.7
Corn	40		
Post-extraction soybean meal (toasted)	20		
Mineral-vitamin supplement *	40		

* calcium carbonate (20 g/kg), monocalcium phosphate (2 g/kg), sodium chloride (5 g/kg), sodium bicarbonate (25 g/kg), vit. A (8000 IU/kg), vit. D3 (1200 IU/kg), vit. E (25 IU/kg), vit. K (0.4 mg), vit. B1 (0.4 mg), vit. B2 (3.2 mg), vit. B6 (0.4 mg), vit. B12 (12 mg), biotin (80 mg), folic acid (0.45 mg), nicotinic acid (16 mg), pantothenic acid (6 mg).

**Table 2 animals-12-02773-t002:** Average fatty acid content of basic feed, hay and ethyl esters of PUFAs obtained from linseed oil [8].

Fatty Acid	Hay	Feed	Linseed Oil Ethyl Esters	Acid	Hay	Feed	Linseed Oil Ethyl Esters
Saturated fatty acids				Unsaturated fatty acids			
C6:0	0.57	-	-	C14:1	-	0.07	-
C8:0	0.66	0.02	-	C16:1	2.33	0.32	-
C10:0	0.64	-	-	C17:1	-	0.05	-
C12:0	1.26	0.03	-	C18:1	-	-	16.73
C14:0	2.4	0.12	-	C18:2n-6c	16.82	50.17	16.68
C15:0	-	0.04	-	C18:2n-6t	17.66	21.32	-
C16:0	28.83	15.39	4.44	C18:3n-6	2.41	-	-
C17:0	-	0.1	-	C18:3n-3	5.3	5.9	58.71
C18:0	4.98	4.24	3.43	C20:4n-6	-	0.07	-
C20:0	-	0.38	-	C20:5n-3	1.75	-	-
				C22:6n-3	-	0.56	-

**Table 3 animals-12-02773-t003:** Average content of omega-6 and omega-3 acids acids and n-6/n-3 ratio in feed and esters and the ratio of these acids.

	n-6 Acids	n-3 Acids	n-6/n-3
Basic feed	72.03%	6.46%	11.15
Linseed oil ethyl esters	16.68%	58.71%	0.28
Acid content in 5 mL of esters	0.71 g	2.50 g	0.28
Content in feed with added ethyl esters	3.02 g	2.71 g	1.11

**Table 4 animals-12-02773-t004:** Blood count results of white termond rabbits.

	WBC (10^9^/L)	Lym (10^9^/L)	Mono (10^9^/L)	Gran (10^9^/L)	RBC (10^12^/L)	HGB (g/L)	HCT (%)	MCV (fL)	MCH (pg)	MCHC (g/L)	RDW (%)	PLT (10^9^/L)
	Mean ± SD	Mean ± SD	Mean ± SD	Mean ± SD	Mean ± SD	Mean ± SD	Mean ± SD	Mean ± SD	Mean ± SD	Mean ± SD	Mean ± SD	MEAN ± SD
I-L-S C	7.87 ± 1.12	3.43 ± 1.34	0.41 ± 0.12	4.02 ± 0.14	5.48 ± 0.51	108.11 ± 10.96	32.52 ± 2.94	59.49 ± 2.35	19.69 ± 1.04	331.89 ± 4.76	14.38 ± 1.06	313.33 ± 31.26
I-L-S E	8.41 ± 1.36	3.80 ± 0.69	0.36 ± 0.01	4.25 ± 0.89	5.91 ± 0.62	115.86 ± 13.37	34.58 ± 3.39	58.64 ± 1.89	19.55 ± 1.06	334.14 ± 8.78	13.92 ± 0.43	413.38 ± 106.09
II-L-W C	7.39 ± 0.30	3.04 ± 0.92	0.28 ± 0.13	4.04 ± 1.00	6.32 ± 0.30	123.51 ± 7.08	36.59 ± 1.20	59.25 ± 1.87	20.14 ± 0.97	340.00 ± 6.44	13.37 ± 1.17	339.89 ± 25.51
II-L-W E	6.96 ± 1.16	3.07 ± 0.93	0.23 ± 0.09	3.66 ± 0.30	6.11 ± 0.56	121.50 ± 9.09	35.79 ± 2.82	59.00 ± 1.67	19.74 ± 0.54	333.35 ± 8.61	13.42 ± 0.38	345.53 ± 73.29
III-O-S C	8.96 ± 1.75	3.96 ± 0.58	0.45 ± 0.08	3.86 ± 0.88	5.43 ± 0.54	111.66 ± 5.35	33.17 ± 2.19	63.11 ± 1.43	21.79 ± 0.05	341.69 ± 1.45	14.51 ± 0.59	396.08 ± 109.94
III-O-S E	9.10 ± 1.94	5.07 ± 0.23	0.50 ± 0.04	3.91 ± 0.34	5.54 ± 0.74	115.50 ± 11.74	34.21 ± 3.29	64.24 ± 0.77	22.47 ± 0.19	346.88 ± 1.58	14.81 ± 1.09	293.88 ± 102.03
IV-O-W C	7.33 ± 0.99	3.99 ± 0.68	0.44 ± 0.06	3.65 ± 0.50	6.88 ± 0.49	128.36 ± 7.06	36.97 ± 0.63	58.53 ± 2.88	22.04 ± 0.82	338.72 ± 7.02	13.88 ± 0.62	492.65 ± 66.53
IV-O-W E	7.45 ± 0.61	3.98 ± 0.57	0.48 ± 0.03	4.06 ± 0.73	7.11 ± 0.62	131.76 ± 6.86	37.01 ± 1.45	58.44 ± 1.87	22.83 ± 0.72	340.31 ± 4.94	14.71 ± 0.86	499.42 ± 68.81
Additive												
C	7.89 ± 1.20	3.61 ± 0.89	0.40 ± 0.11	3.89 ± 0.63	6.03 ± 0.75	117.91 ± 10.97	34.81 ± 2.66	60.10 ± 2.64	20.92 ± 1.27	338.08 ± 6.01	14.04 ± 0.90	385.49 ± 119.43
E	7.98 ± 1.45	3.98 ± 0.93	0.39 ± 0.12	3.97 ± 0.57	6.17 ± 0.82	121.15 ± 11.32	35.40 ± 2.68	60.08 ± 2.87	21.15 ± 1.69	338.67 ± 8.05	14.21 ± 0.88	388.05 ± 120.64
Condition												
L	7.66 ± 1.07	3.34 ^a^ ± 0.91	0.32 ^A^ ± 0.11	3.99 ± 0.63	5.96 ± 0.54	117.25 ± 10.84	34.87 ± 2.82	59.10 ^a^ ± 1.70	19.78 ^A^ ± 0.82	334.84 ^a^ ± 7.04	13.77 ± 0.83	353.03 ± 81.78
O	8.21 ± 1.49	4.25 ^b^ ± 0.68	0.47 ^B^ ± 0.05	3.87 ± 0.57	6.24 ± 0.95	121.82 ± 11.21	35.34 ± 2.53	61.08 ^b^ ± 3.18	22.28 ^B^ ± 0.63	341.90 ^b^ ± 4.94	14.48 ± 0.79	420.51 ± 140.19
Season												
S	8.58 ^a^ ± 1.44	4.07 ± 0.94	0.43 ^a^ ± 0.08	4.01 ± 0.58	5.59 ^A^ ± 0.56	112.78 ^A^ ± 9.77	33.62 ^A^ ± 2.69	61.37 ^A^ ± 2.86	20.88 ± 1.48	338.65 ± 7.61	14.40 ± 0.79	354.17 ± 131.61
W	7.28 ^b^ ± 0.74	3.52 ± 0.83	0.36 ^b^ ± 0.13	3.85 ± 0.62	6.61 ^B^ ± 0.60	126.28 ^B^ ± 7.71	36.59 ^B^ ± 1.56	58.81 ^B^ ± 1.85	21.19 ± 1.50	338.10 ± 6.56	13.85 ± 0.89	419.37 ± 95.82
*p*-value												
Additive	0.8622	0.2694	0.9604	0.7821	0.5565	0.4063	0.5657	0.9851	0.4700	0.8137	0.6060	0.9546
Condition	0.2993	0.0128	0.0004	0.6613	0.2324	0.2470	0.6439	0.0228	0.0000	0.0115	0.0537	0.1481
Season	0.0225	0.1125	0.0414	0.5694	0.0004	0.0027	0.0088	0.0050	0.3339	0.8255	0.1179	0.1613
Interaction	0.6464	0.5610	0.9209	0.3916	0.4248	0.5492	0.6474	0.5745	0.7666	0.5993	0.9903	0.2689

Experimental factor: Additive—addition of linseed oil ethyl esters (control—C, experimental—E), Condition—animal living conditions (laboratory—L, outdoor cage—O), Season—season of experiment (summer—S, winter—W), Interaction—interaction between factors; 4 stages of experiment: I-L-S (I-Laboratory—Summer), II-L-W (II-Laboratory—Winter), III-O-S (III-Outdoor—Summer) and IV-O-W (IV-Outdoor—Winter); ^A, B^—highly significant differences at the level of *p* < 0.01; ^a, b^—significant differences at the level of *p* < 0.05.

**Table 5 animals-12-02773-t005:** Biochemical parameters of the blood of termond white rabbits.

	ALT (IU/L)	AST (IU/L)	Fibrinogen (g/L)	Urea (mmol/L)
	Mean ± SD	Mean ± SD	Mean ± SD	Mean ± SD
I-L-S C	34.87 ± 9.25	19.65 ± 5.96	3.85 ± 0.53	3.95 ± 0.48
I-L-S E	44.42 ± 7.57	23.28 ± 3.63	3.48 ± 0.48	3.59 ± 0.37
II-L-W C	42.42 ± 10.17	21.70 ± 5.14	3.68 ± 0.60	6.54 ± 1.98
II-L-W E	46.60 ± 6.56	24.47 ± 6.95	3.74 ± 0.53	4.26 ± 1.10
III-O-S C	52.51 ± 6.71	26.73 ± 1.24	4.22 ± 0.39	7.14 ± 1.30
III-O-S E	52.84 ± 3.54	24.97 ± 1.51	3.95 ± 0.18	6.55 ± 0.41
IV-O-W C	38.32 ± 3.67	17.53 ± 2.00	4.82 ± 0.57	5.06 ± 0.89
IV-O-W E	39.55 ± 1.84	17.76 ± 1.26	4.72 ± 0.63	4.62 ± 0.25
Additive				
C	42.03 ± 9.63	21.41 ± 4.99	4.14 ± 0.64	5.67 ^a^ ± 1.71
E	45.85 ± 6.78	22.62 ± 4.57	3.97 ± 0.64	4.76 ^b^ ± 1.27
Condition				
L	42.08 ± 8.60	22.28 ± 5.10	3.69 ^A^ ± 0.48	4.59 ^A^ ± 1.57
O	45.81 ± 8.08	21.75 ± 4.53	4.43 ^B^ ± 0.55	5.84 ^B^ ± 1.29
Season				
S	46.16 ± 9.76	23.66 ± 4.12	3.88 ± 0.45	5.31 ± 1.75
W	41.72 ± 6.39	20.37 ± 4.87	4.24 ± 0.75	5.12 ± 1.38
*p*-value				
Additive	0.1839	0.4743	0.4227	0.0421
Condition	0.1941	0.7553	0.0025	0.0078
Season	0.1266	0.0652	0.0964	0.6531
Interaction	0.5773	0.6742	0.7631	0.2277

Experimental factor: Additive—addition of linseed oil ethyl esters (control—C, or experimental—E), Condition—animal living conditions (laboratory—L, or outdoor cage—O), Season—season of experiment (summer—S, or winter—W), Interaction—interaction between factors; 4 stages of experiment: I-L-S (I-Laboratory—Summer), II-L-W (II-Laboratory—Winter), III-O-S (III-Outdoor—Summer) and IV-O-W (IV-Outdoor—Winter); ^A, B^—highly significant differences at the level of *p* < 0.01; ^a, b^—significant differences at the level of *p* < 0.05.

**Table 6 animals-12-02773-t006:** Fatty acid profile of rabbit blood erythrocytes.

	SFA	UFA	MUFA	PUFA	n-3	n-6	n-6/n-3
	Mean ± SD	Mean ± SD	Mean ± SD	Mean ± SD	Mean ± SD	Mean ± SD	Mean ± SD
I-L-S C	39.28 ± 0.33	56.54 ± 0.14	20.83 ± 0.15	35.70 ± 0.24	7.95 ± 0.23	27.12 ± 0.10	3.42 ± 0.10
I-L-S E	37.63 ± 2.57	60.25 ± 3.16	21.70 ± 0.77	38.55 ± 2.42	9.53 ± 1.29	28.34 ± 1.14	3.00 ± 0.29
II-L-W C	39.26 ± 0.49	56.31 ± 0.37	21.04 ± 0.22	35.46 ± 0.22	7.49 ± 0.18	27.40 ± 0.05	3.67 ± 0.08
II-L-W E	37.87 ± 0.78	59.81 ± 2.56	21.36 ± 0.32	38.46 ± 2.24	9.39 ± 1.61	28.45 ± 0.60	3.09 ± 0.51
III-O-S C	36.69 ± 0.49	56.24 ± 0.47	21.13 ± 0.19	35.10 ± 0.30	7.10 ± 0.11	27.39 ± 0.24	3.86 ± 0.06
III-O-S E	35.10 ± 1.54	61.35 ± 4.00	21.65 ± 0.51	39.70 ± 3.49	9.87 ± 2.13	29.10 ± 1.21	3.04 ± 0.60
IV-O-W C	37.21 ± 0.04	56.55 ± 0.15	20.98 ± 0.06	35.57 ± 0.10	7.27 ± 0.07	27.71 ± 0.05	3.81 ± 0.03
IV-O-W E	35.89 ± 1.14	60.90 ± 3.74	21.22 ± 0.21	39.68 ± 3.53	10.22 ± 2.48	28.77 ± 0.98	2.94 ± 0.72
Additive							
C	38.11 ^A^ ± 1.27	56.41 ^A^ ± 0.31	21.00 ^A^ ± 0.18	35.46 ^A^ ± 0.30	7.45 ^A^ ± 0.36	27.40 ^A^ ± 0.25	3.69 ^A^ ± 0.19
E	36.62 ^B^ ± 1.86	60.58 ^B^ ± 2.97	21.48 ^B^ ± 0.47	39.10 ^B^ ± 2.62	9.75 ^B^ ± 1.68	28.67 ^B^ ± 0.91	3.02 ^B^ ± 0.47
Condition							
L	38.51 ^A^ ± 1.42	58.23 ± 2.57	21.23 ± 0.51	37.04 ± 2.08	8.59 ± 1.28	27.83 ± 0.82	3.30 ± 0.38
O	36.22 ^B^ ± 1.19	58.76 ± 3.41	21.25 ± 0.36	37.51 ± 3.11	8.61 ± 2.05	28.24 ± 1.00	3.41 ± 0.60
Season							
S	37.18 ± 2.05	58.59 ± 3.20	21.33 ± 0.56	37.26 ± 2.71	8.61 ± 1.59	27.99 ± 1.09	3.33 ± 0.46
W	37.56 ± 1.41	58.39 ± 2.85	21.15 ± 0.25	37.29 ± 2.61	8.59 ± 1.82	28.08 ± 0.76	3.38 ± 0.54
*p*-value							
Additive	0.0077	0.0007	0.0038	0.0007	0.0008	0.0006	0.0007
Condition	0.0003	0.5985	0.9206	0.5581	0.9661	0.1418	0.4480
Season	0.4491	0.8415	0.2432	0.9726	0.9685	0.7334	0.7659
Interaction	0.9960	0.8898	0.6745	0.8560	0.9568	0.6921	0.8742

Experimental factor: Additive—addition of linseed oil ethyl esters (control—C, or experimental—E), Condition—animal living conditions (laboratory—L, or outdoor cage—O), Season—season of experiment (summer—S, or winter—W), Interaction—interaction between factors; 4 stages of experiment: I-L-S (I-Laboratory—Summer), II-L-W (II-Laboratory—Winter), III-O-S (III-Outdoor—Summer) and IV-O-W (IV-Outdoor—Winter); ^A, B^—highly significant differences at the level of *p* < 0.01.

**Table 7 animals-12-02773-t007:** Content of omega 3 and omega 6 fatty acids in rabbit blood erythrocytes.

.	Omega 3 Acids					Omega 6 Acids					
	ALA	DHA	EPA	OTA	DPA	LA	GLA	AA	EDA	DGLA	DTA
	Mean ± SD	Mean ± SD	Mean ± SD	Mean ± SD	Mean ± SD	Mean ± SD	Mean ± SD	Mean ± SD	Mean ± SD	Mean ± SD	Mean ± SD
I-L-S C	0.77 ± 0.02	3.28 ± 0.08	2.69 ± 0.16	0.02 ± 0.00	1.20 ± 0.04	22.39 ± 0.10	2.01 ± 0.04	1.19 ± 0.03	0.32 ± 0.01	0.73 ± 0.01	0.49 ± 0.02
I-L-S E	1.15 ± 0.29	4.32 ± 0.69	2.69 ± 0.22	0.02 ± 0.00	1.36 ± 0.11	23.17 ± 0.77	2.20 ± 0.17	1.40 ± 0.16	0.36 ± 0.04	0.72 ± 0.03	0.50 ± 0.01
II-L-W C	0.69 ± 0.03	3.20 ± 0.07	2.46 ± 0.13	0.02 ± 0.00	1.12 ± 0.03	22.92 ± 0.05	1.82 ± 0.04	1.15 ± 0.01	0.33 ± 0.01	0.69 ± 0.01	0.49 ± 0.01
II-L-W E	1.18 ± 0.38	4.12 ± 0.87	2.87 ± 0.31	0.02 ± 0.00	1.21 ± 0.06	23.52 ± 0.35	1.94 ± 0.03	1.40 ± 0.16	0.35 ± 0.02	0.74 ± 0.03	0.49 ± 0.01
III-O-S C	0.35 ± 0.01	3.23 ± 0.04	2.25 ± 0.08	0.02 ± 0.00	1.27 ± 0.07	22.55 ± 0.15	2.39 ± 0.13	0.96 ± 0.08	0.27 ± 0.02	0.82 ± 0.03	0.41 ± 0.02
III-O-S E	0.86 ± 0.25	4.50 ± 1.07	3.13 ± 0.52	0.02 ± 0.00	1.36 ± 0.09	23.62 ± 0.86	2.81 ± 0.18	1.13 ± 0.12	0.30 ± 0.01	0.84 ± 0.04	0.44 ± 0.03
IV-O-W C	0.42 ± 0.01	3.26 ± 0.02	2.36 ± 0.05	0.02 ± 0.00	1.21 ± 0.01	22.70 ± 0.01	2.13 ± 0.03	1.16 ± 0.02	0.31 ± 0.01	0.97 ± 0.01	0.45 ± 0.01
IV-O-W E	0.93 ± 0.44	4.77 ± 1.30	3.17 ± 0.67	0.25 ± 0.02	1.18 ± 0.25	23.41 ± 0.67	2.32 ± 0.11	1.25 ± 0.13	0.32 ± 0.03	1.02 ± 0.04	0.46 ± 0.02
Additive											
C	0.56 ^A^ ± 0.18	3.24 ^A^ ± 0.06	2.44 ^A^ ± 0.19	0.02 ± 0.00	1.20 ± 0.07	22.64 ^A^ ± 0.22	2.09 ^A^ ± 0.22	1.12 ^A^ ± 0.10	0.03 ± 0.01	0.80 ± 0.11	0.46 ± 0.04
E	1.03 ^B^ ± 0.38	4.43 ^B^ ± 0.90	2.97 ^B^ ± 0.45	0.08 ± 0.02	1.28 ± 0.16	23.43 ^B^ ± 0.61	2.32 ^B^ ± 0.35	1.30 ^B^ ± 0.17	0.03 ± 0.01	0.83 ± 0.13	0.47 ± 0.03
Condition											
L	0.95 ^a^ ± 0.31	3.73 ± 0.70	2.68 ± 0.24	0.02 ± 0.00	1.22 ± 0.11	23.00 ± 0.57	1.99 ^A^ ± 0.16	1.28 ^A^ ± 0.16	0.02 ^A^ ± 0.01	0.72 ^A^ ± 0.03	0.49 ^A^ ± 0.01
O	0.64 ^b^ ± 0.38	3.94 ± 1.03	2.73 ± 0.57	0.08 ± 0.02	1.26 ± 0.14	23.07 ± 0.67	2.41 ^B^ ± 0.28	1.13 ^B^ ± 0.14	0.03 ^B^ ± 0.01	0.91 ^B^ ± 0.09	0.44 ^B^ ± 0.03
Season											
S	0.78 ± 0.38	3.83 ± 0.82	2.69 ± 0.41	0.02 ± 0.00	1.30 ^a^ ± 0.10	22.93 ± 0.72	2.35 ^A^ ± 0.33	1.17 ± 0.19	0.04 ± 0.01	0.78 ^A^ ± 0.06	0.46 ± 0.04
W	0.81 ± 0.38	3.84 ± 0.95	2.72 ± 0.47	0.08 ± 0.02	1.18 ^b^ ± 0.12	23.14 ± 0.48	2.05 ^B^ ± 0.20	1.24 ± 0.14	0.02 ± 0.01	0.86 ^B^ ± 0.15	0.47 ± 0.02
*p*-value											
Additive	0.0009	0.0009	0.0015	0.3373	0.0857	0.0011	0.0001	0.0007	0.0554	0.2070	0.1426
Condition	0.0166	0.4784	0.7046	0.3373	0.4895	0.7403	0.0000	0.0023	0.0000	0.0000	0.0000
Season	0.8310	0.9820	0.8494	0.3373	0.0143	0.3190	0.0000	0.1115	0.0758	0.0028	0.2009
Interaction	0.8199	0.7659	0.3981	0.3373	0.7882	0.8218	0.3822	0.4841	0.6702	0.3655	0.7168

Experimental factor: Additive—addition of linseed oil ethyl esters (control—C, or experimental—E), Condition—animal living conditions (laboratory—L, or outdoor cage—O), Season—season of experiment (summer—S, or winter—W), Interaction—interaction between factors; 4 stages of experiment: I-L-S (I-Laboratory—Summer), II-L-W (II-Laboratory—Winter), III-O-S (III-Outdoor—Summer) and IV-O-W (IV-Outdoor—Winter); ^A, B^—highly significant differences at the level of *p* < 0.01; ^a, b^—significant differences at the level of *p* < 0.05.

**Table 8 animals-12-02773-t008:** Fatty acid profile in rabbit blood serum.

	SFA	UFA	MUFA	PUFA	n-3	n-6	n-6/n-3
	Mean ± SD	Mean ± SD	Mean ± SD	Mean ± SD	Mean ± SD	Mean ± SD	Mean ± SD
I-L-S C	34.64 ± 0.24	58.21 ± 1.01	25.54 ± 0.35	33.00 ± 0.30	1.84 ± 0.05	30.81 ± 0.28	16.87 ± 0.34
I-L-S E	35.86 ± 1.36	60.06 ± 1.50	25.61 ± 0.04	34.45 ± 1.45	2.42 ± 0.61	31.36 ± 0.32	13.03 ± 4.48
II-L-W C	34.73 ± 0.25	57.91 ± 0.78	25.46 ± 0.29	31.79 ± 0.21	1.77 ± 0.03	29.39 ± 0.23	16.66 ± 0.28
II-L-W E	35.22 ± 0.47	57.82 ± 0.93	24.74 ± 0.78	33.08 ± 1.71	2.62 ± 1.04	29.86 ± 0.68	12.58 ± 4.08
III-O-S C	34.48 ± 0.73	58.53 ± 0.38	25.55 ± 0.40	32.98 ± 0.16	1.80 ± 0.04	30.69 ± 0.14	17.12 ± 0.36
III-O-S E	34.25 ± 0.65	60.64 ± 2.32	26.15 ± 0.17	34.50 ± 2.18	2.94 ± 1.04	30.95 ± 1.06	11.53 ± 4.25
IV-O-W C	34.74 ± 0.09	58.03 ± 0.17	25.52 ± 0.12	32.50 ± 0.13	1.90 ± 0.01	30.12 ± 0.13	15.94 ± 0.10
IV-O-W E	34.57 ± 0.87	60.67 ± 2.60	26.05 ± 0.50	34.64 ± 2.11	2.90 ± 1.01	31.18 ± 1.04	11.67 ± 4.04
Additive							
C	34.65 ± 0.37	58.17 ^a^ ± 0.62	25.52 ± 0.26	32.57 ^A^ ± 0.54	1.83 ^A^ ± 0.06	30.25 ^a^ ± 0.61	16.65 ^A^ ± 0.52
E	34.98 ± 1.01	59.80 ^b^ ± 2.07	25.64 ± 0.71	34.17 ^B^ ± 1.74	2.72 ^B^ ± 0.83	30.84 ^b^ ± 0.94	12.20 ^B^ ± 3.66
Condition							
L	35.11 ± 0.81	58.50 ± 1.33	25.34 ^A^ ± 0.53	33.08 ± 1.38	2.16 ± 0.64	30.35 ± 0.88	14.78 ± 3.32
O	34.51 ± 0.59	59.47 ± 1.95	25.82 ^B^ ± 0.41	33.66 ± 1.62	2.38 ± 0.84	30.74 ± 0.76	14.07 ± 3.62
Season							
S	34.81 ± 0.97	59.36 ± 1.65	25.71 ± 0.36	33.73 ± 1.37	2.25 ± 0.71	30.95 ^A^ ± 0.56	14.64 ± 3.66
W	34.81 ± 0.51	58.61 ± 1.75	25.44 ± 0.64	33.00 ± 1.60	2.30 ± 0.79	30.14 ^B^ ± 0.87	14.21 ± 3.31
*p*-value							
Additive	0.2660	0.0148	0.4725	0.0099	0.0046	0.0310	0.0022
Condition	0.0518	0.1232	0.0091	0.3064	0.4252	0.1407	0.5652
Season	0.9863	0.2245	0.1181	0.2020	0.8683	0.0045	0.7302
Interaction	0.4948	0.3151	0.2807	0.7263	0.7112	0.3834	0.7514

Experimental factor: Additive—addition of linseed oil ethyl esters (control—C, or experimental—E), Condition—animal living conditions (laboratory—L, or outdoor cage—O), Season—season of experiment (summer—S, or winter—W), Interaction—interaction between factors; 4 stages of experiment: I-L-S (I-Laboratory—Summer), II-L-W (II-Laboratory—Winter), III-O-S (III-Outdoor—Summer) and IV-O-W (IV-Outdoor—Winter); ^A, B^—highly significant differences at the level of *p* < 0.01; ^a, b^—significant differences at the level of *p* < 0.05.

**Table 9 animals-12-02773-t009:** Content of omega 3 and omega 6 fatty acids in rabbit blood serum.

	Omega 3 Acids					Omega 6 Acids					
	ALA	DHA	EPA	OTA	DPA	LA	GLA	AA	EDA	DGLA	DTA
	Mean ± SD	Mean ± SD	Mean ± SD	Mean ± SD	Mean ± SD	Mean ± SD	Mean ± SD	Mean ± SD	Mean ± SD	Mean ± SD	Mean ± SD
I-L-S C	0.51 ± 0.01	0.23 ± 0.03	0.06 ± 0.01	0.02 ± 0.01	1.03 ± 0.03	25.39 ± 0.24	2.30 ± 0.17	1.74 ± 0.05	0.15 ± 0.00	0.69 ± 0.01	0.53 ± 0.01
I-L-S E	0.65 ± 0.17	0.65 ± 0.06	0.08 ± 0.02	0.01 ± 0.01	1.32 ± 0.32	25.86 ± 0.18	2.42 ± 0.31	1.69 ± 0.16	0.18 ± 0.03	0.71 ± 0.01	0.50 ± 0.03
II-L-W C	0.51 ± 0.01	0.20 ± 0.01	0.08 ± 0.02	0.02 ± 0.01	0.97 ± 0.03	24.53 ± 0.20	2.30 ± 0.03	1.28 ± 0.04	0.15 ± 0.00	0.64 ± 0.01	0.50 ± 0.01
II-L-W E	0.61 ± 0.19	0.59 ± 0.06	0.10 ± 0.02	0.01 ± 0.01	1.31 ± 0.30	24.75 ± 0.31	2.46 ± 0.30	1.33 ± 0.04	0.17 ± 0.02	0.67 ± 0.03	0.49 ± 0.01
III-O-S C	0.50 ± 0.03	0.23 ± 0.01	0.07 ± 0.01	0.01 ± 0.00	1.00 ± 0.02	25.68 ± 0.17	2.38 ± 0.02	1.28 ± 0.04	0.15 ± 0.01	0.69 ± 0.01	0.52 ± 0.01
III-O-S E	0.72 ± 0.23	0.80 ± 0.51	0.09 ± 0.03	0.01 ± 0.01	1.02 ± 0.81	25.79 ± 0.65	2.53 ± 0.34	1.29 ± 0.10	0.16 ± 0.02	0.70 ± 0.02	0.51 ± 0.02
IV-O-W C	0.51 ± 0.00	0.21 ± 0.01	0.06 ± 0.00	0.02 ± 0.00	1.10 ± 0.02	25.01 ± 0.09	2.56 ± 0.06	1.19 ± 0.04	0.16 ± 0.01	0.69 ± 0.01	0.51 ± 0.01
IV-O-W E	0.75 ± 0.26	0.75 ± 0.48	0.09 ± 0.03	0.02 ± 0.00	1.30 ± 0.25	25.70 ± 0.74	2.80 ± 0.18	1.27 ± 0.10	0.17 ± 0.01	0.75 ± 0.04	0.49 ± 0.01
Additive											
C	0.51 ^a^ ± 0.02	0.22 ^A^ ± 0.02	0.07 ^A^ ± 0.01	0.02 ± 0.01	1.02 ± 0.06	25.15 ^b^ ± 0.48	2.39 ± 0.13	1.37 ± 0.23	0.15 ^A^ ± 0.01	0.68 ^A^ ± 0.02	0.51 ^A^ ± 0.01
E	0.68 ^b^ ± 0.19	0.70 ^B^ ± 0.48	0.09 ^B^ ± 0.02	0.02 ± 0.01	1.24 ± 0.43	25.53 ^a^ ± 0.65	2.55 ± 0.29	1.39 ± 0.20	0.17 ^B^ ± 0.02	0.71 ^B^ ± 0.04	0.50 ^B^ ± 0.02
Condition											
L	0.57 ± 0.13	0.42 ± 0.42	0.08 ± 0.02	0.02 ± 0.01	1.16 ± 0.25	25.13 ^b^ ± 0.59	2.37 ^b^ ± 0.21	1.51 ^A^ ± 0.23	0.16 ± 0.02	0.68 ^A^ ± 0.03	0.50 ± 0.02
O	0.62 ± 0.19	0.50 ± 0.42	0.08 ± 0.02	0.02 ± 0.01	1.11 ± 0.38	25.54 ^a^ ± 0.54	2.57 ^a^ ± 0.23	1.26 ^B^ ± 0.08	0.16 ± 0.01	0.71 ^B^ ± 0.03	0.51 ± 0.01
Season											
S	0.60 ± 0.16	0.48 ± 0.43	0.07 ± 0.02	0.01 ^a^ ± 0.00	1.09 ± 0.40	25.68 ^A^ ± 0.37	2.41 ± 0.23	1.50 ^A^ ± 0.24	0.16 ± 0.02	0.70 ± 0.01	0.51 ^a^ ± 0.02
W	0.60 ± 0.17	0.44 ± 0.41	0.08 ± 0.02	0.02 ^b^ ± 0.00	1.17 ± 0.22	25.00 ^B^ ± 0.58	2.53 ± 0.24	1.27 ^B^ ± 0.07	0.16 ± 0.01	0.69 ± 0.05	0.50 ^b^ ± 0.01
*p*-value											
Additive	0.0123	0.0078	0.0065	0.6607	0.1398	0.0328	0.0764	0.5467	0.0044	0.0082	0.0065
Condition	0.4515	0.6286	0.8259	0.6607	0.7083	0.0203	0.0399	0.0000	0.9999	0.0035	0.4669
Season	0.9791	0.8203	0.11985	0.0399	0.5898	0.0006	0.1923	0.0000	0.3791	0.1785	0.0120
Interaction	0.8341	0.9918	0.5119	0.6607	0.8351	0.2127	0.8882	0.7899	0.9999	0.1785	0.1985

Experimental factor: Additive—addition of linseed oil ethyl esters (control—C, or experimental—E), Condition—animal living conditions (laboratory—L, or outdoor cage—O), Season—season of experiment (summer—S, or winter—W), Interaction—interaction between factors; 4 stages of experiment: I-L-S (I-Laboratory—Summer), II-L-W (II-Laboratory—Winter), III-O-S (III-Outdoor—Summer) and IV-O-W (IV-Outdoor—Winter); ^A, B^—highly significant differences at the level of *p* < 0.01; ^a, b^—significant differences at the level of *p* < 0.05.

## Data Availability

The data presented in this study are available in this article.

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
