# Peer review of "The Effect of the Housing System, Season and the Linseed Oil Ethyl Esters Additive on Selected Blood Parameters in Rabbits"

_animals, 2022, doi:10.3390/ani12202773_

Round 1
Reviewer 1 Report
The Effect of the Housing System, Season and the Linseed Oil Ethyl Esters Additive on Selected Blood Parameters of Rabbits
Dear Authors,
the problem raised in the manuscript is very interesting because it describes the possibilities of enriching dietary rabbit meat additionally with unsaturated fatty acids, the source of which can be linseed oil used in the diet of these animals, which then improves the health-promoting properties of this end product. Additionally, the use of linseed oil is associated as an experimental factor with two animal keeping systems and two seasons. As a result of the use of linseed oil, there was a significant increase (at least p <0.05) in the meat of the level of UFA, PUFA, n-3 and n-6 fatty acids. But as usually no one from us is perfect, and below I add some recommendations, comments and questions helpful in correction of the text before publication:
Line 47-51
P-value should be added in text
Line 52
One space must be deleted
Line 75
One space need to be deleted
Line 82 and 87
Repetition of information in introduction. Please delete one from doubled paragraphs or merge them and rewrite de novo.
Line 116
‘Arrangements of experience” please change for Arrangements of experiment or Experimental design
Line 123
Full description of abbreviations in this place will be needed
Line 126
In manuscript is: ‘In all stages of the experiment, 16 males of termond rabbits were used, which were 125 randomly divided into two groups: control (C) without additive and experimental (E) 126 with addition of linseed oil ethyl esters, 8 in each’, maybe better will be to write control (C-without addition), experimental (E- with addition of linseed oil ethyl esters), 8 replication in each.
Line 133
Nutrition instead of Feeding
Line 134
Maybe requirements will be better word than feeding standards
Line 136
Please only check if feed was granulated or pelleted
Line 149
In text is experience, please change for experiment
Line 157
In table 2, header in first column, there is ‘Acid’, please add full form: Fatty acid
Line 196
Mean values instead of average values will be more proper in my opinion
Line 197
Please delete one space after and/before linseed oil ethyl…
Line 197-198
Information about what kind of ANOVA was used in experiment is needed: one-way, three way (because in tables interaction between three factors is described)
Line 198
Statistica 13.3 (TIBCO Software Inc., Palo Alto, CA, USA) could be added to the References in position [22] and then change is needed in case following publications 23-36.
Line 202
In Results during describing significance level information about values for estimators of analysed variable will be useful during to read the text, ie. like in line 206: ‘The living conditions of the rabbits significantly (p<0,01) influenced the content of monocytes’ in animals kept in laboratory and outdoor cages (respectively 0,47 x 109/L and 0,32 x 109/L).
Line 204
‘The addition of linseed oil ethyl esters did not affect’ P-value needed (p>0,05)
Line 215
One and three way ANOVA is presented in a table, but there is lack of some information. P-value for first part of the table (one way ANOVA arrangement) in second part of table three factorial design is used and data for it are presented in experimental design (2x2x2), but there is only one overall interaction presented between factor (there is lack of interactions between two factors: addition*conditions, addition*season, conditions*season. Or only explanation why only one overall interaction is presented
Line 226, 240, 256, 267 and 283
The same like in line 215
Line 288
Maybe it is better to merge information in first four sentences of paragraph, because article Melilo (2007) [22] appears 3 times in a row.
Line 378
DOI or ISSN or http:// information needed on the end of article/publication
Line 391, 401, 408, 409 and 411
Better is to change publications/articles with Polish titles on English title with annotation [In Polish] on the end before DOI or ISSN or http:// information
Author Response
Replies to reviewer (animals-1922591)
The authors would like to thank the reviewer for all comments and suggestions which may improve manuscript quality. The authors made changes to the text in line with the reviewers' renewed comments.
The authors decided to present only overall interaction between factor (excluding interactions between two factors: addition*conditions, addition*season, conditions*season) in order not to enter too much data that could be unreadable and confusing for the reader.
Following the reviewer's suggestions, information has been added, which should improve the readability of the abstract. The fragments of the manuscript specified by the reviewer were rewritten or merged to improve the readability of the information.
The DOI informations were added according to Instructions for Authors (MDPI).
Every effort has been made to improve the quality of the manuscript. The authors hope that the corrections made in line with the reviewers' suggestions have improved the readability and quality of the manuscript.
Yours faithfully,
Authors
Reviewer 2 Report
all tables and specific in table 4: explain the letters as I-L-S C blow the table please which must be agreement with divided groups in methodology.
table 5: re statistical analysis not show significant letter between LWC and LWE ?
in general the justification and hypothesis is not building on the strong data, you can clear it more and re written the aim of study
Author Response
Replies to reviewer (animals-1922591)
The authors would like to thank the reviewer for all comments and suggestions which may improve manuscript quality. The authors made changes to the text in line with the reviewers' renewed comments.
The tables represent each stage of experiment (I-L-S, I-L-W, II-L-S etc.) and group of experiment (C - control, E- experimental), which was explained in the Methodology chapter. However, following the reviewer's suggestions, explanations of the letters have been added under each table, which should improve the table readability.
Despite the visible differences in the values ​​of the LWC and LWC data (Table 5), the statistical analysis did not show any significant differences. This is probably due to the higher standard deviation for the LWC group.
Every effort has been made to improve the quality of the manuscript; as suggested by reviewers, some parts of the manuscript were rewritten. The authors hope that the corrections made in line with the reviewers' suggestions have improved the readability and quality of the manuscript.
Yours faithfully,
Authors
Reviewer 3 Report
In this article, the authors investigate the effects of environmental conditions and season on blood parameters in rabbits. They also experiment the effects of dietary addition of linseed oil ethyl esters on blood parameters in rabbits. They show that the housing conditions and season affected morphological and biochemical parameters of rabbit blood. Furthermore, they confirmed that the linseed oil ethyl ester supplement used did not adversely affect the health of the rabbits and produced expected beneficial changes in the fatty acid profile in erythrocytes and blood serum.
The topic addressed is important for interpreting the rabbit hematology result, but I think this manuscript needs to be considered more sufficiently before publication.
Ÿ Materials and Methods
1. Please organize chapter numbers. (There are no number 2-3 and 2-6.)
2. Line 123-124. Please show what “L” and “O” indicate.
Ÿ Discussion
3. Please more discuss the significance of results obtained in this study, especially in relation to fatty acid profile.
Author Response
Replies to reviewer (animals-1922591)
The authors would like to thank the reviewer for all comments and suggestions which may improve manuscript quality. The authors made changes to the text in line with the reviewers' renewed comments.
Following the reviewer's suggestions, explanations of the letters L and O have been added. The fragments of the manuscript specified by the reviewer were rewritten or merged to improve the readability of the information.
The authors hope that the corrections made in line with the reviewers' suggestions have improved the readability and quality of the manuscript.
Yours faithfully,
Authors
Round 2
Reviewer 2 Report
thank you for all revisions
good luck